# In Vitro Anticancer Activity of Two Ferrocene-Containing Camphor Sulfonamides as Promising Agents against Lung Cancer Cells

**DOI:** 10.3390/biomedicines10061353

**Published:** 2022-06-08

**Authors:** Maria Schröder, Maria Petrova, Zlatina Vlahova, Georgi M. Dobrikov, Ivaylo Slavchev, Evdokia Pasheva, Iva Ugrinova

**Affiliations:** 1Institute of Molecular Biology “Akad. Roumen Tsanev”, Bulgarian Academy of Sciences, Acad. G. Bonchev Str, bl 21, 1113 Sofia, Bulgaria; marias82@abv.bg (M.S.); mhristova84@abv.bg (M.P.); vlahova94@gmail.com (Z.V.); eva@bio21.bas.bg (E.P.); 2Institute of Organic Chemistry with Center of Phytochemistry, Bulgarian Academy of Sciences, Acad. G. Bonchev Str, bl 9, 1113 Sofia, Bulgaria; gmdob@orgchm.bas.bg (G.M.D.); ivailo.slavchev@orgchm.bas.bg (I.S.)

**Keywords:** ferrocene derivatives, lung cancer, cytotoxicity, apoptosis, autophagy, ROS

## Abstract

The successful design of antitumour drugs often combines in one molecule different biologically active subunits that can affect various regulatory pathways in the cell and thus achieve higher efficacy. Two ferrocene derivatives, DK-164 and CC-78, with different residues were tested for cytotoxic potential on non-small lung cancer cell lines, A549 and H1299, and non-cancerous MRC5. DK-164 demonstrated remarkable selectivity toward cancer cells and more pronounced cytotoxicity against A549. The cytotoxicity of CC-78 toward H1299 was even higher than that of the well-established anticancer drugs cisplatin and tamoxifen, but it did not reveal any noticeable selective effect. DK-164 showed predominantly pro-apoptotic activity in non-small cell lung carcinoma (NSCLC) cells, while CC-78 caused accidental cell death with features characteristic of necrosis. The level of induced autophagy was similar for both substances in cancer cells. DK-164 treatment of A549, H1299, and MRC5 cells for 48 h significantly increased the fluorescence signal of the NFkB (nuclear factor ‘kappa-light-chain-enhancer’ of activated B-cells) protein in the nucleus in all three cell lines, while CC-78 did not provoke NFkB translocation in any of the tested cell lines. Both compounds caused a significant transfer of the p53 protein in the nucleus of A549 cells but not in non-cancerous MRC5 cells. In A549, DK-164 generated oxidative stress close to the positive control after 48 h, while CC-78 had a moderate effect on the cellular redox status. In the non-cancerous cells, MRC5, both compounds produced ROS similar to the positive control for the same incubation period. The different results related to the cytotoxic potential of DK-164 and CC-78 associated with the examined cellular mechanisms induced in lung cancer cells might be used to conclude the specific functions of the various functional groups in the ferrocene compounds, which can offer new perspectives for the design of antitumour drugs.

## 1. Introduction

Chemotherapy plays a significant role in the treatment of cancer worldwide. That is why the identification and study of new promising molecular structures as potential anticancer drugs is a great challenge. Chemotherapeutics, combining several biologically active subunits in one molecule, can modulate different regulatory pathways in the cell and thus achieve higher efficacy than drugs that affect only one cellular process [1,2,3]. The success of cisplatin in antitumour therapy as one of the main chemotherapeutics in clinical practice motivated scientists’ interest in metal compounds. In the last few decades, several cisplatin analogues have been developed as potential antitumour agents, but only two of them, carboplatin and oxaliplatin, have reached the clinical application [4]. Side effects, such as nephrotoxicity, neurotoxicity, gametogenesis, lack of specific action, and the development of resistance, limit the use of platinum derivatives [5]. To overcome these problems and to improve patient survival, research focuses on the discovery and design of drugs with new chemical structures and mechanisms of action.

Organometallic compounds show several advantages as potential candidates for anticancer treatment [6]. They are known for showing considerable diversity in their structures and for exhibiting variable oxidation states, while also being kinetically stable; their character is relatively lipophilic; the metal cation exhibits redox properties. They are also able to bind biological target molecules, and ligands for them can be designed rationally [7]. The metallocenes’ antitumour effects are most often linked to their ability to form metallocene–DNA complexes, to inhibit DNA and RNA synthesis and/or to induce an increase in free radical levels in cancer cells [8]. Ferrocenes are metallocenes, i.e., organometallic compounds consisting of two cyclopentadienyl rings bound to a central iron atom [9]. Organometallic complexes differ in structure and chemical characteristics from well-known platinum-based anticancer drugs.

Non-small cell lung cancer is the most common type of lung cancer and accounts for about 80–85% of cases. As a rule, non-small cell lung cancer originates from different types of epithelial cells in the lung. In general, non-small cell neoplasms grow and develop more slowly than small ones. However, they are less sensitive to chemotherapy and radiation therapy than small cell lung cancer. The molecular basis of lung cancer is complex and heterogeneous. It is among the tumours with the highest number of reported genetic aberrations [10].

Today, cancer is often still looked upon as a global and homogenous disease; the tumours themselves are often assumed as a homogenous population of cells. However, it has become clear that during tumourigenesis, malignancies become highly heterogenous, creating a mixed population of cells that exhibit different molecular characteristics and, consequentially, demonstrate different therapy responses [11]. Therefore, a more thorough understanding of the complex phenomena associated with tumourigenesis, as well as of the exact mechanism of action of various anticancer drugs, is essential for the development of personalised and effective therapeutic concepts.

In our previous investigations, a newly synthesised ferrocene-containing camphor sulfonamide DK-164 was tested on two breast cancer and one breast non-cancer cell lines and showed promising anticancer activity [12] (see Appendix A). However, its main disadvantage was the poor solubility in aqueous media. In this study, we presented a new analogue of DK-164, possessing better water solubility at physiological pH values.

Here, we report the cytotoxic and antineoplastic potential of the two ferrocene-containing camphor sulfonamides on two lung cancer cell lines, A549 and H1299, and one non-cancerous lung cell line, MRC5. The analyses of the cell cycle, apoptosis, and autophagy provide the first information about the mechanism of action of both substances on lung cells. Furthermore, we tested the effect of the compounds on the cellular localisation of NFkB and the tumour suppressor factor p53, critical players in essential cellular processes and for the induction of reactive oxygen species (ROS).

## 2. Materials and Methods

### 2.1. Chemistry

All details concerning chemistry, synthetic methods, analyses, and spectra of synthesised compounds are presented in Appendix A.

### 2.2. Cell Culture

Human lung carcinoma cell lines, A549 and H1299, and non-cancerous lung fibroblasts, MRC5, were purchased from the ATTC collection of cell cultures. A549 and H1299 cells were cultured in complete media consisting of F12K or RPMI-1640, respectively, modified with 10% fetal bovine serum (Thermo Fisher Scientific, Waltham, MA, USA), 100 units/mL penicillin, and 100 µg/mL streptomycin (Thermo Scientific). MRC5 cells were cultured in ATCC-formulated Eagle’s Minimum Essential Medium modified with 10% Fetal bovine serum (Thermo Fisher Scientific, Waltham, MA, USA), 100 units/mL penicillin, and 100 µg/mL streptomycin (Thermo Fisher Scientific, Waltham, MA, USA). Cells were grown in 5% CO_2_ in an air-humidified incubator at 37 °C. For all experiments, only cells growing in the exponential phase were used.

### 2.3. In Vitro Cytotoxicity Test

The cytotoxicity of the compounds DK-164 and CC-78 was evaluated on two human lung cancer cell lines—A549 and H1299—and the human lung fibroblast cell line MRC5 via an MTT dye reduction assay, as previously described [13,14]. For positive controls, we used cisplatin and tamoxifen—agents that induce inhibition of cellular proliferation. The cells were seeded at 3 × 10^3^ per well in 96-well flat-bottom plates and pre-incubated for 24 h (at 37 °C and 5% CO_2_). For the treatment, all used compounds (DK-164, CC-78, cisplatin, and tamoxifen) were applied in serial dilutions ranging from 4 µM to 512 µM. The final concentration of DMSO at the highest drug concentration in the growth medium was 0.5%. The control group of cells received a medium containing only 0.5% DMSO. The cells were incubated for 72 h. MTT was performed according to the manufacturer’s protocol and as already described [12]. Absorbance was measured on an ELISA plate reader Varioscan Lux (Thermo Fisher Scientific, Waltham, MA, USA) with a test wavelength of 570 nm. IC50 values were calculated in GraphPad Sigma v.8.0 (Dotmatics, San Diego, CA, USA) software using the “log of concentration vs normalized response (variable slope)” algorithm. 

### 2.4. Flow Cytometry

An Annexin V Apoptosis Detection Kit FITC/ PI (Thermo Fisher Scientific, Waltham, MA, USA) was used to analyse apoptosis by flow cytometry, following the manufacturer’s instructions. The cells were plated in 12-well plates and were treated with the calculated IC50 concentrations of DK-164 or CC-78 (see Table 1) for 24 and 48 h. For the analysis, the cells were resuspended in Annexin V Binding Buffer (AVBB) at a concentration of ~1 × 10^6^ cells/mL in 100 µL, and 5 µL of Annexin V-FITC was added. After 15 min of incubation, the samples were washed with AVBB, resuspended in 200 µL AVBB containing PI, and incubated on ice in the dark for 30 min. The labelled cells were analysed by flow cytometry with proper compensating settings on a Becton Dickinson FACScalibur instrument (BD Biosciences, San Jose, CA, USA). The percentages of live, early apoptotic, late apoptotic, and necrotic cells were quantified with FlowJo v.10.8.1 software (BD Biosciences, Ashland, OR, USA).

### 2.5. Immunofluorescence Microscopy

To observe the autophagic marker, LC3, as well as the dynamics of the transcription factors NFkB and p53, cells were grown on glass coverslips, fixed in 3.7% paraformaldehyde in PBS for 5 min at RT, and then permeabilised with 0.1% Triton X-100 in PBS for 5 min. After blocking in 10% fetal calf serum, 1% BSA, and 0.1% TX-100 in PBS, the coverslips were incubated in primary antibodies (polyclonal rabbit anti-LC3 (Abcam, Cambridge, UK), monoclonal mouse anti-p53 (Biolegend, San Diego, CA, USA), polyclonal rabbit anti-NFkB (Abcam), and monoclonal mouse anti-vimentin (Biolegend, San Diego, CA, USA) diluted in PBS (1:100–1:500) at 37 °C for 30 min. For visualisation, secondary antibodies, the donkey anti-mouse Alexa 555 antibody (Thermo Fisher Scientific, Waltham, MA, USA) and goat anti-rabbit Alexa 488 antibody (Thermo Scientific) at 1:2000 dilution, were used. The coverslips were mounted in ProLong Diamond mounting media (Thermo Fisher Scientific, Waltham, MA, USA) containing 400 ng/mL DAPI. Images and consequent image analysis were performed as previously described [12].

### 2.6. Reactive Oxygen Species (ROS) Analysis

ROS were detected using a 2′,7′-dichlorodihydrofluorescein diacetate (DCF-DA) ROS Assay Kit from OZBiosciences (Marseille, France) according to the manufacturer’s instructions. Briefly, A549 and MRC5 cells were plated in 96-well black plates, optimised for plate readers, at 2000 cells/well and 4500 cells/well, respectively, and incubated for 24 h. Then the cells were exposed to IC25 concentrations of DK-164 or CC-78 for 24 and 48 h, and *tert*-Butyl hydroperoxide (TBHP) was used as a positive control. In preliminary experiments, the appropriate concentrations of TBHP for each cell line were determined as follows: 45 µM for A549 and 5 µM for MRC5. Afterwards, the cells were stained with 25 μM DCF-DA at 37 °C for 30 min in the dark. Subsequently, the extracellular dye was discarded, and the cells were rinsed with cold PBS twice. Positive controls were incubated with TBHP for 3 h before adding the fluorogenic dye DCF-DA. The experiment was repeated three times, each performed in triplicate. ROS levels were measured at Exc/Em 485/535 nm using a VarioscanLux Plate Reader (Thermo Fisher Scientific, Waltham, MA, USA). The fluorescent intensity was analysed by SkanIt v.6.0.1 software (Thermo Fisher Scientific, Waltham, MA, USA).

### 2.7. JC-1 Mitochondrial Membrane Potential Assay

A549 and MRC5 cells were seeded in 4-well glass slide chambers (Lab-Tek) and treated for 24 h with IC25 concentrations of DK-164 and CC-78. Then the cells were washed once with 1xPBS and incubated for 30 min with 5 µg/mL of 5,5′,6,6′-tetrachloro-1,1′,3,3′-tetraethylbenzimidazolcarbocyanine iodide (JC-1, Sigma-Aldrich, Burlington, MA, USA) in culture medium at 37 °C. The cells were washed with a growth medium and then observed and imaged by a Zeiss AxioVert 200M microscope (Carl Zeiss, Oberkochen, Germany) using a 100× objective lens, equipped with a CCD camera AxioCam MRm. The ratio of red/green fluorescence intensity was analysed by ImageJ v1.8.0_172 software (Madison, WI, USA).

### 2.8. Statistical Analysis

For statistical analysis, GraphPad Prism v.8.0 software (Dotmatics, San Diego, CA, USA) was used. Most data are presented as mean values ± SD of three independent experiments. Statistical differences in the microscopic analysis considering autophagy and the translocation of p53 and NFkB were calculated by a one-way single factor ANOVA test. For statistical analysis of the results concerning ROS detection and JC-1 assay, Dunnett’s multiple comparisons test in ordinary two-way ANOVA was used. A *p*-value < 0.05 was considered statistically significant. * *p* < 0.05, ** *p* < 0.01, *** *p* < 0.005, and **** *p* < 0.001.

## 3. Results

### 3.1. Chemistry

One disadvantage of metal–organic compounds is their pronounced hydrophobicity and poor solubility in a physiological environment. The previously characterised ferrocene sulfonamide, DK-164 [12] (See Appendix A), showed promising antitumour activity and selectivity on breast cancer cell lines but problematic water solubility. To improve the solubility of DK-164, a series of synthetic modifications were performed to structurally optimise the compound in which increased hydrophilicity was achieved by substituting functional groups (Figure 1).

Our synthetic strategy was based on synthesising a small series of camphor sulfonamides, **8**–**13**, through reactions between 1*S*-camphor sulfonylchloride (**1**) and reagents **2**–**7**. These reagents were selected to introduce in compounds **8**–**13** different moieties, which may increase their water solubility at physiological pH values (pH 5–7). The synthesis of **8**–**13** (except **9**) was performed in good yields in dry dichloromethane using triethylamine as a base. Due to the insolubility of reagent **3** in dichloromethane, compound **9** was synthesised in excess of dry pyridine (serving as a solvent base). Sulfonamides **8**–**13** were tested for solubility in water at pH 5–7. Unfortunately, only compound **12** demonstrated noticeable solubility at pH 5–6. Thus, sulfonamide **12** was chosen for further derivatisation with ferrocene carbaldehyde (**14**). The reaction of **12** with **14** was performed in dry DMSO using KOBu-*t* as a base, leading to desired analogues of DK-164, namely target compound **12a** in high yield. All compounds in this study were new. They were obtained in pure form by column chromatography or recrystallisation and characterised by NMR, MS, and melting points (for detailed analytical data of the compounds, see Appendix A).

### 3.2. Solubility

The new derivatives were subjected to in vitro cytotoxicity tests, and the sulfonamide CC-78 (**12a**), with high solubility and the most pronounced antiproliferative activity, was selected for further studies. DK-164 and CC-78 have different residues conjugated to the sulfonamide moiety, which affects the solubility of the compounds. In DK-164, the sulfonamide functionality was bound to an aliphatic and pH-indifferent *tert*-butyl group, which determines the hydrophobic nature of the molecule. In contrast, CC-78 contains two basic nitrogen atoms, able to form water-soluble salts even with weak organic acids in human cells (see Appendix A).

### 3.3. In Vitro Evaluation of DK-164 and CC-78 Cytotoxicity

The cytotoxic profile of DK-164 and CC-78 was tested on two NSCLC cell lines with different invasive potential, A549 considered with better prognosis and H1299 with higher invasive potential and negative outcome, and on non-cancerous MRC5 cells (Figure 2).

The summarised IC50 values are presented in (Table 1). DK-164 demonstrated remarkable selectivity toward cancer cells and more pronounced cytotoxicity for A549. An interesting finding was that the cytotoxicity of CC-78 toward H1299 was noticeably higher compared to well-established anticancer drugs, such as cisplatin and tamoxifen. CC-78 showed better general cytotoxic potential, but it did not reveal any distinguishable selective effect.

### 3.4. Study of Apoptosis in Lung Cells Treated with DK-164 and CC-78

In the next step, we analysed the potential of DK-164 and CC-78 to induce apoptosis in A549, H1299 cancer cells, and non-cancerous MRC5. To distinguish early from late apoptosis and necrosis, the AnnexineV-FITC/PI method was used. The cells were treated for 24 h and 48 h with the calculated IC50 concentrations of each compound and analysed by flow cytometry. Representative dot plots of these experiments are shown in Figure 3.

FACS analyses showed that each substance caused apoptosis and necrosis in all three lung cell lines, but with different intensities and time dependencies. DK-164 showed predominantly pro-apoptotic activity in NSCLC cells, which was registered at a later stage of treatment and mostly affected A549 cells with functional p53 proteins. After 48 h treatment, the early apoptotic cells were increased approximately 55 times compared to the untreated controls. In H1299, treatment resulted in a five-times increase in necrotic cells after 48 h. Non-cancerous MRC5 cells responded to treatment as early as 24 h, with a five-times increase in apoptosis. The number of viable cells remained stable, and the initial apoptotic cells showed signs of necrosis.

All experiments demonstrated a significant reduction in viable cells after treatment with CC-78, and unlike with DK-164, the dominant mechanism of cell death was necrosis. Particularly affected were H1299 cells, in which necrotic cells increased 14.7 times at 24 h and even 22 times at 48 h of treatment. A considerable increase in the number of necrotic cells was also observed in MRC5, and the effect was registered at the 24th hour without intensifying with prolonged treatment. Rapid action of CC-78 within 24 h was also measured in A549 cells, in which the necrotic fraction increased 6.8-fold without potentiating the effect at 48 h. The results were plotted and are presented in Figure 4.

### 3.5. Autophagy in Lung Cells Induced by Treatment with DK-164 and CC-78

Many studies suggest that autophagy plays an essential role in some diseases, although it is unclear whether it has a protective or inhibitory function [15]. Some of the main characteristics of autophagy explain its dual role in cancer [16,17].

To study the level of autophagy in A549 and H1299 lines, as well as in non-cancerous MRC5 cells caused by DK-164 and CC-78 for 24 h, we performed immunocytochemistry to monitor LC3 puncta formation. To visualise the cellular shape, vimentin was used, and nuclei were co-stained with DAPI (Figure 5A). The data for statistics were obtained by quantifying the observed immunofluorescence signal using ImageJ software for image analysis. The graphs were generated by GraphPad Prism v.8.0 software (Dotmatics, San Diego, CA, USA) (Figure 5B).

DK-164 enhanced the autophagy signals in all three lung cell lines, leading to an increase in the number of cells with active autophagy. In control, untreated cells, the LC3 signal was comparatively dim, diffused, and evenly distributed in the cytoplasm. In the LC3-positive cells, characteristic dot-shaped fluorescent spots (LC3 puncta) were observed, marking the formed LC3–II-phosphatidylethanolamine complexes. DK-164 caused an approximately 2.8-fold increase in the fraction of LC3-positive cells in A549 and H1299 tumour cells and 3.3-fold in MRC5 cells. CC-78 increased the percentage of cells with active autophagy by about 2.5-fold in all tested cell lines (Figure 5B). Higher levels of endogenous autophagy were observed in untreated NSCLC cells compared to MRC5 cells. High basal levels of autophagy are characteristic of several cancers with mutations in RAS proteins [18], such as H1299 (with mutant NRAS proteins) and A549 (with mutant KRAS proteins) cells.

### 3.6. The Effect of DK-164 and CC-78 on the Cellular Localisation of the Regulatory Proteins NF-kB and p53 in Lung Cells

The resistance of tumour cells to apoptosis is one of the reasons for the failure of antitumour therapy with cytotoxic drugs or radiation and for the poor prognosis of patients [19].

The p53 protein is an essential mediator of endogenous apoptosis, which motivated us to monitor the dynamics of the p53 protein. A549 and MRC5 cells were treated with IC50 concentrations of DK-164 and CC-78 for 48 h, and cover slides were incubated with antibodies against p53. Nuclei were co-stained with DAPI. Because H1299 cells do not express the p53 protein, they were excluded from the experiment. The observations are presented in Figure 6.

DK-164 and CC-78 caused the translocation of the p53 protein in the nucleus of A549 cells but not in non-cancerous MRC5 cells. The nuclear signal intensity at A549 increased approximately 1.4-fold after 48 h of treatment with both substances. The similar effect of DK-164 and CC-78 on p53 localisation indicated that the chemical modifications made in CC-78 did not affect the examined feature.

NFκB is a specific transcription factor from the Rel protein family that occurs in every cell and is particularly characteristic in B-lymphocytes. Research has shown that tumour cells are “addicted” to the activated form of NFκB [20], and it is essential to clarify to what extent new substances affect this process.

To monitor the dynamics of the NFκB protein, A549, H1299, and MRC5 cells were treated with IC50 concentrations of DK-164 and CC-78. After 48 h of treatment, cover slides were incubated with antibodies against NFκB. Nuclei were co-stained with DAPI. The observations are presented in Figure 7.

DK-164 treatment of A549, H1299, and MRC5 cells for 48 h increased the fluorescence signal of the NFκB protein in the nucleus by 1.4 times in all three cell lines. Of particular interest is the result for the modified compound CC-78, which did not lead to the accumulation of NFκB in the nucleus in any of the three cell lines.

### 3.7. Activation of Reactive Oxygen Species (ROS) in Lung Cancer Cells by DK-164, CC-78

One promising therapeutic target of cancer treatment is the modulation of reactive oxygen species (ROS) production. In tumour cells, ROS levels are elevated due to various features, such as high metabolic activity, oncogene activation, hypoxia conditions, and loss of p53 protein function [21]. A549 and MRC5 cells were exposed to IC25 concentrations of DK-164 or CC-78 for 24 and 48 h. The ROS generation was detected by 2′,7′-dichlorodihydrofluorescein diacetate (DCF-DA). TBHP was used as a positive control. The results are presented in Figure 8. In A549 cells after 24 h of incubation with CC-78, a slightly higher ROS level was observed compared to those in non-treated cells. The treatment with DK-164 practically showed no effect at 24 h. However, after 48 h of treatment, DK-164 generated oxidative stress close to that of the positive control.

In MRC5 cells, the oxidative stress induced by DK-164 was higher than that caused by CC-78 after 24 h of treatment. The ROS level after 48 h of treatment with both compounds was comparable to those in the positive control. (Figure 8).

To further illustrate the changes in cellular redox balance that occured under the influence of the newly synthesised ferrocene derivatives, DK-164 and CC-78, the accompanying changes in the mitochondrial membrane potential were also monitored. The principle of visualisation is based on the property of JC-1, a cationic carbocyanine dye (green), that after accumulation in healthy mitochondria, forms aggregates (red) [22]. Any event that breaks the potential of the mitochondrial membrane prevents the accumulation of JC-1 in the mitochondria. Thus, the dye is dispersed back to the cytoplasm, transiting from red fluorescence (JC-1 aggregates) to a diffused green signal (JC-1 monomers) (Figure 9A). In A549 cells, the reduction in the red fluorescent signal versus the green one was observed after treatment with both tested substances, whereas in MRC5 cells, DK-164 had a more prominent effect (Figure 9B).

## 4. Discussion

One disadvantage of metal–organic compounds is their pronounced hydrophobicity and poor solubility in the physiological environment. Recently in our group, we characterised the newly synthesised ferrocene-containing camphor sulphonamide DK-164 as a compound with promising antitumour potential, tested on breast cancer cells [12]. The main disadvantage of this compound was its poor solubility under physiological conditions. To overcome this issue, we performed a series of structural modifications and selected a new promising candidate—CC-78.

The alteration of biological activity due to chemical modifications is a common occurrence. In this study, we demonstrated an increase in the cytotoxic effect and loss of selectivity after replacing the *tert*-butyl group of DK-164 with piperazine in CC-78, which contributed to better solubility and bioavailability at physiological pH values.

The examined ferrocene-containing camphor sulfonamides DK-164 and CC-78 demonstrated different cytotoxic selectivity toward lung cancer cell lines A549 and H1299 compared to the non-cancerous MRC5. In concern with selectivity, molecular docking studies have been made with many sulfonamide derivatives [23]. It was shown that the sulfonamide tail of the active compounds could interact with Zn^2+^ in the active site of the enzyme carbonic anhydrase IX/XII, suggesting the importance of the sulfonamide moiety in inhibiting its action [24]. In this regard, selective CA IX/XII inhibitors with antiproliferative effects enhanced in hypoxia have been developed as a strategy for selective action in tumour cells. In most sulfonamide CA inhibitors, the sulfonamide group is terminal and can bind freely to the active site of the enzyme. We can speculate that DK-164 and CC-78 have different binding capacities to enzyme pockets due to their different steric features. The more voluminous piperazine in CC-78 could sterically prevent such binding of the sulfonamide moiety and make it challenging for this mechanism to achieve a selective effect.

In our hands, DK-164 showed predominantly pro-apoptotic activity in NSCLC cells, which is already reported for sulphonamides, while CC-78 caused accidental cell death with features characteristic of necrosis. It is already described that some ferrocene derivatives cause a rapid rounding of cells, followed by their separation from the cell monolayer. The action of the ammonium chain as a detergent in physiological pH, reported by Lu et al., may explain the rapid onset of necrosis in all cells [25,26]. Such protonation is also possible at the piperazine tail of CC-78. The mechanism of inactivating tumour cells or cell death (ageing, apoptosis, or necrosis) varies depending on the structure of the ferrocene complexes, the used cell lines, and the doses administered [27]. The biological activity of CC-78 is most likely influenced to a greater extent by the properties of the ferrocene moiety and the protonation of additional nitrogen atoms. In addition to ferrocene, DK-164 and CC-78 contain sulfonamide and camphor; each of these subunits provides new opportunities for modulating cell signalling pathways and for their synergistic interaction.

One of the established characteristics of ferrocifenes (well-known conjugated ferrocene derivatives, active against breast cancer [27]) is that at concentrations close to IC50, they inhibit cell growth by ageing rather than cell death by apoptosis due to stimulating the secretion of IL-8 and TNF-α in cancer cell models characterised by different sensitivity to pro-apoptotic stimuli (MDA-MB-231, MCF-7, U373, Hs683, and B16F10) [28,29]. The secretion of these cytokines may be related to the activity of the AP-1 transcription factor. For other ferrocene conjugates, apoptotic cell death has been successfully observed, as shown by morphological analyses, activation of effector caspases 3 and 7, increased DNA fragmentation, phosphatidylserine (PS) externalisation, and the necrosis process at the highest concentration [30]. Since compounds DK-164 and CC-78 contain conjugated ferrocene moieties, they could be considered analogues of ferrocifenes. In our case, CC-78 exhibited a clear tendency to induce cell death characterised by membrane permeabilisation, while DK-164 more likely stimulated apoptotic processes.

In this study, we showed that both compounds increased the population of cells with activated autophagy in all tested cell lines. This effect was more pronounced in cancer cells A549 and H1299. Anticancer treatments induce autophagy in cells as a cytoprotective response to reduce cellular stress caused by toxic action [31]. It is generally accepted that autophagy plays a protective role in cancer cells against chemotherapy, so integrating autophagy inhibitors into anticancer therapy is considered a promising therapeutic strategy [32]. Conversely, increased induction of autophagy under the influence of anticancer drugs or inducers of autophagy also leads to the death of cancer cells [33].

On the other hand, anticancer therapy can stimulate autophagic pathways, which mediate so-called autophagic cell death (ACD). By definition, ACD is a mode of cell death inhibited by a specific blockage of the autophagic pathway [34]. It is known that the mechanism of action of several chemotherapeutic drugs involves the activation of autophagy [35,36,37]. In their work with glioblastoma cells, Würstle et al. demonstrated that the DNA alkylating agent temozolomide can induce ACD in an EGFR-independent manner [38]. Quite interesting are the findings for the natural BH3 mimetic gossypol (a natural phenol metabolically formed via isoprenoid pathway and derived from the cotton plant genus Gossypium [39]), which can induce autophagic cell death in apoptosis-resistant cancer cells [40,41] and, at the same time, leads to apoptotic cell death in cells with intact apoptotic pathways [42,43]. The research group of Salazar et al. found that the major active ingredient in cannabinoids, such as Tetrahydrocannabinol (THC), can act as a stimulus for ACD in hepatocellular carcinoma and glioblastoma [44], but does not show similar cytotoxicity to normal non-malignant cells [45].

Regarding the effect on the translocation of p53 proteins in the nucleus of A549 cells but not in non-cancerous MRC5 cells, both compounds DK-164 and CC-78 act similarly. Whether or not the p53-activating action of DK-164 and CC-78 is associated with blocking its negative regulators may be the subject of more in-depth research to accurately classify these compounds as part of therapeutic concepts combining various approaches to suppress tumourigenesis.

The tested DK-164 and CC-78 showed striking differences regarding their effect on NFκB translocation tested on lung cancer and non-cancerous cells. While DK-164 induced the transfer of the transcription factor from the cytoplasm to the nucleus, CC-78 had no effect. The importance of NFκB activation in cancer treatment is not yet sufficiently elucidated due to various hypotheses about the role of this protein in carcinogenesis. Its activation is generally considered the cause of tumour cell resistance to apoptosis, chemotherapy, and radiation therapy [46]. Other data highlight its pro-apoptotic action as a direct regulator of Fas receptors and Fas-associated apoptosis [47]. In various studies, treatment of tumour cells with UV rays or chemical agents, such as doxorubicin [48] and TN 16 [49], resulted in the activation of NFκB and its pro-apoptotic functions. Numerous different data suggest that the role of NFκB in antitumour therapy varies from case to case and depends on cell type and the characteristics of the apoptotic stimulus [50]. We can speculate that the effect of CC-78 on the transcription factor NFκB coincides with the expected cellular answer to chemotherapeutic stress without causing further activation of NFκB in tumour cells.

As the accumulation of oxidative stress is considered a key mechanism of antitumor activity of ferrocene derivatives [51], the extent to which the studied substances DK-164 and CC-78 modulate the production of reactive oxygen species in A549 and MRC5 cells were also examined. The production of ROS in lung tumour cells upon treatment with DK-164 and CC-78 indicated a different way of action. In A549 lung cancer cells, DK-164 generated oxidative stress close to the positive control after 48 h; in comparison, CC-78 had a moderate effect. It was not the case for the non-cancerous MRC5 lung cells, where, for the same period of incubation, both compounds produced the same ROS levels as the positive control.

There is piling evidence that antitumor agents exert their effects by inducing ROS and reducing GSH levels [52,53]. The imbalance between antioxidants and pro-oxidants results in oxidative stress that may promote cell death [54]. The cancer cells escape programmed cell death regardless of the persistently higher ROS in a more efficient manner than normal cells due to the higher levels of reduced glutathione, which might cause tumour survival.

Concerning DK-164, the observed depolarisation of the mitochondrial membrane coincides with the studied ROS accumulation and NFκB activation and might be considered an indicator of the occurrence of apoptosis, as found by the flow cytometric analysis. In contrast, CC-78 provokes accidental cell death with the feature characteristics of necrosis, which does not seem to involve NFκB activation, and a considerable modulation of the cellular redox balance.

## 5. Conclusions

Our results showed different biological activities of two compounds, DK-164 and CC-78, with the same fundamental organometallic scaffold containing ferrocene, camphor, and sulphonamide. These two compounds differ only by a single substitution of one residue *tert*-butyl group, with another one being piperazine, which led to the loss of essential properties and manifested others. On this matter, our results might provide perspectives or ideas for the rational design of new ferrocene-based anticancer drugs.

## Figures and Tables

**Figure 1 biomedicines-10-01353-f001:**
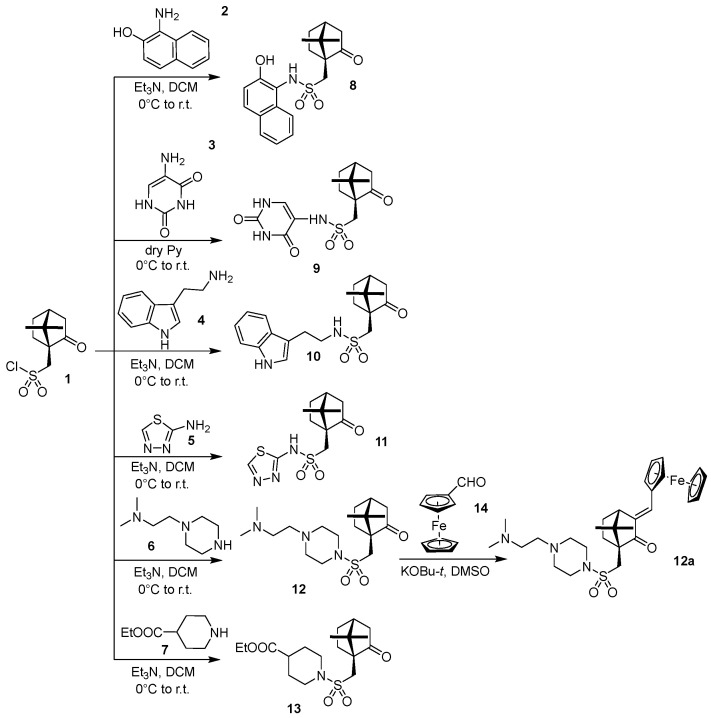
Synthesis of a series of new camphor sulfonamides. Compound **12a** (1*S*,4*S*)-3-((*E*)-ferrocenylmethylidene)-1-(4-(4-(2-(2-(dimethylamino)ethyl)piperazin-1-yl)sulfonyl)methyl)-7,7-dimethylbicyclo[2.2.1]heptan-2-one (working title CC-78) was chosen for further investigations.

**Figure 2 biomedicines-10-01353-f002:**
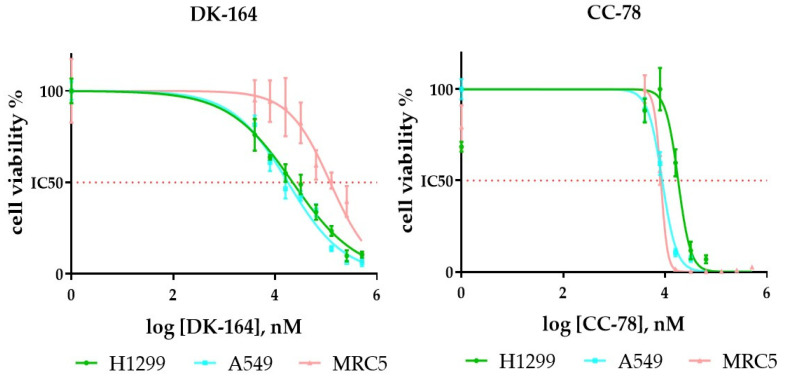
Dose–response curves were used to generate IC50 for the tested compounds, DK-164 and CC-78. The curves were generated by GraphPad Prism v.8.0 software (Dotmatics, San Diego, CA, USA). The data represent mean ± SD (*n* = 4).

**Figure 3 biomedicines-10-01353-f003:**
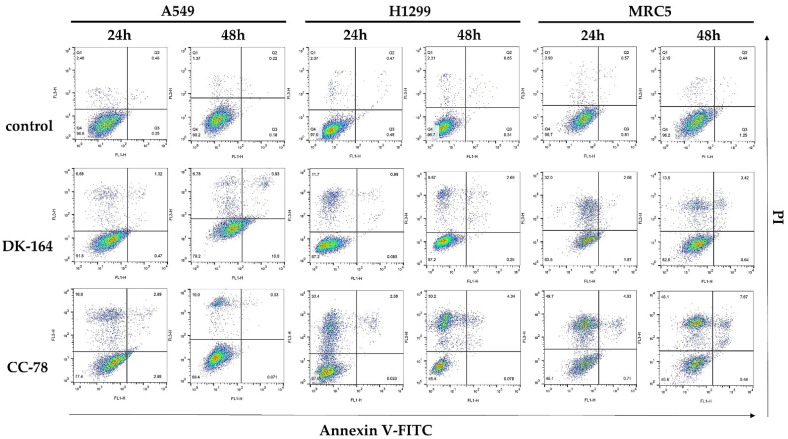
Apoptosis was assessed by staining with Annexin V-FITC/ PI followed by flow cytometry. Representative statistics of this analysis are presented. A549, H1299, and MRC5 cells were treated for 24 h or 48 h with the calculated IC50 concentrations of the compounds DK-164 and CC-78 (see Table 1).

**Figure 4 biomedicines-10-01353-f004:**
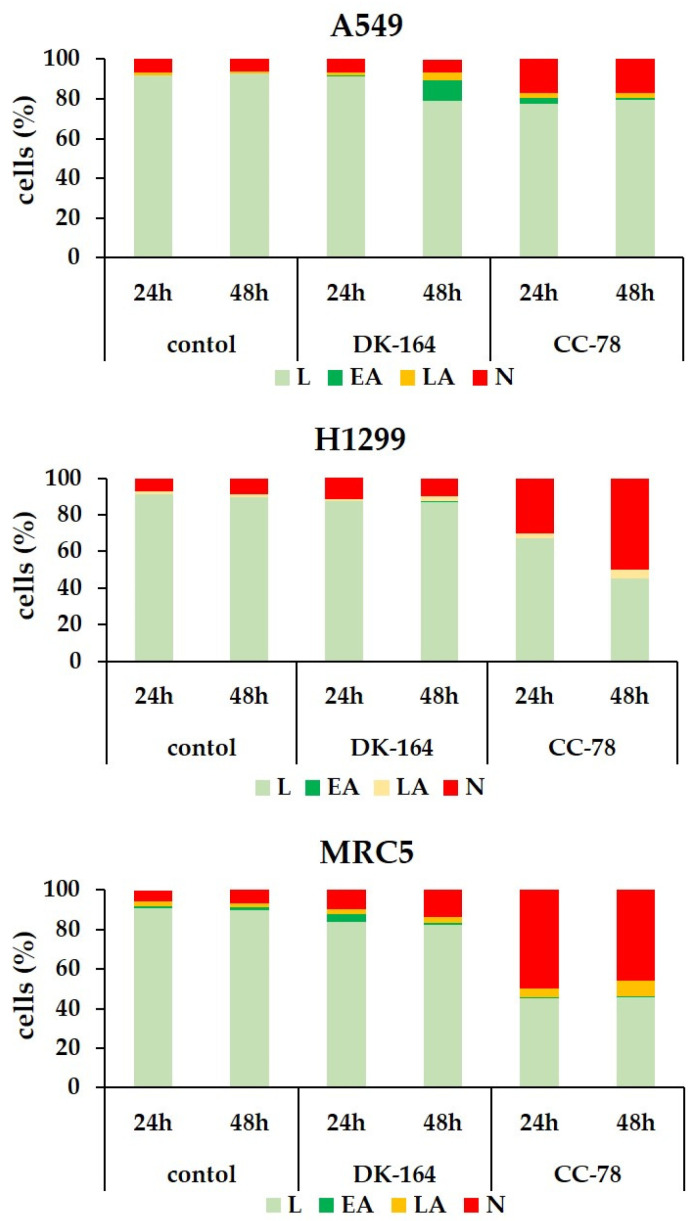
Results of FACS analysis of apoptosis. Rate distribution of living (L, light green), early apoptotic (EA, green), late apoptotic (LA, orange), and necrotic cells (N, red) in A549, H1299, and MRC5 cells treated for 24 h or 48 h with the calculated IC50 concentrations (see Table 1) of the compounds DK-164 and CC-78. All data shown are reproducible and representative of three independent experiments.

**Figure 5 biomedicines-10-01353-f005:**
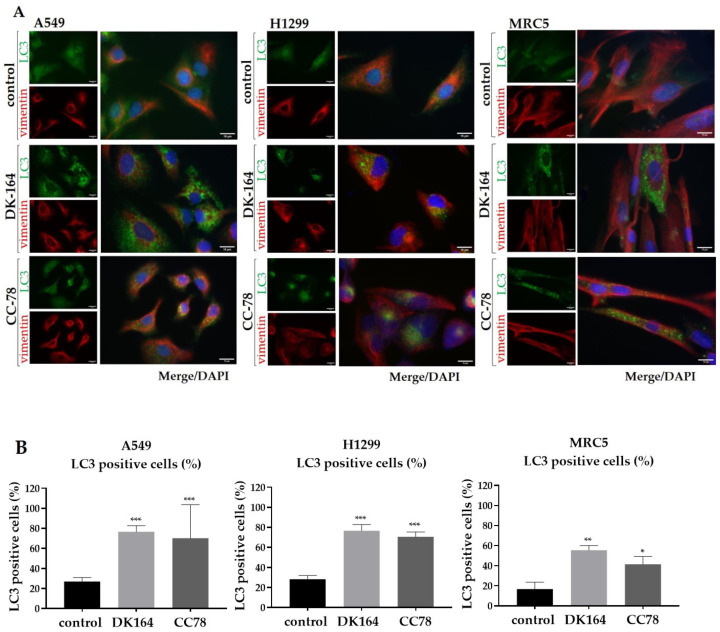
Detection of the autophagy marker LC3 by immunofluorescence. (**A**) Representative immunofluorescent images of A549, H1299, and MRC5 control cells and cells treated with the calculated IC50 concentrations of DK-164 and CC-78 for 24 h and incubated with anti-LC3 antibody (green) and anti-vimentin antibody (red). DNA was co-stained with DAPI (blue). (**B**) Quantitative determination of the fraction of LC3-positive cells was performed using the ImageJ quantification tool. A *p*-value < 0.05 was considered statistically significant, * *p* < 0.05, ** *p* < 0.01, and *** *p* < 0.005. Quantification is based on three independent experiments with >50 cells scored for each condition. Error bars represent standard deviation (SD). The scale bar corresponds to 15 μm.

**Figure 6 biomedicines-10-01353-f006:**
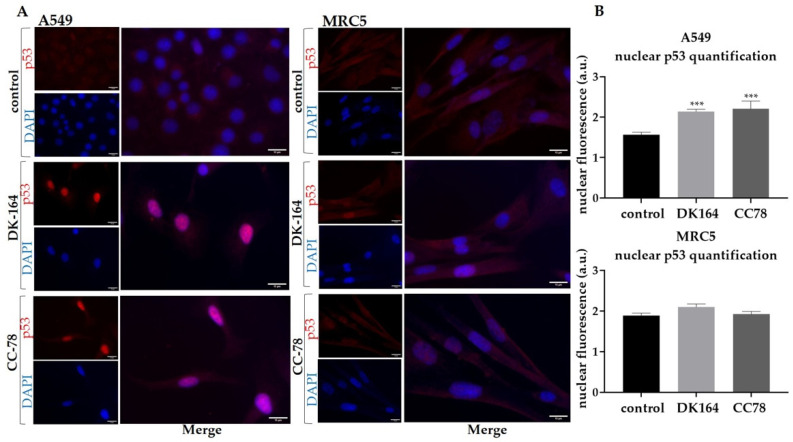
Activation and nuclear translocation of the p53 protein in A549 and MRC5 cells. (**A**) Representative immunofluorescence images of A549 and MRC5 control cells and cells treated with the calculated IC50 concentrations of DK-164 or CC-78 for 48 h and labelled with anti-p53 antibody (red). DNA was co-stained with DAPI (blue). (**B**) Quantitative determination of the intensity of the p53 protein was performed by CellProfiler cell image analysis software and GraphPad Prism 8.0. A *p*-value < 0.05 was considered statistically significant, *** *p* < 0.005. Quantification is based on three independent experiments with >50 cells scored for each condition. Error bars represent the standard deviation (SD). a.u.: arbitrary unit. The scale bar corresponds to 15 μm.

**Figure 7 biomedicines-10-01353-f007:**
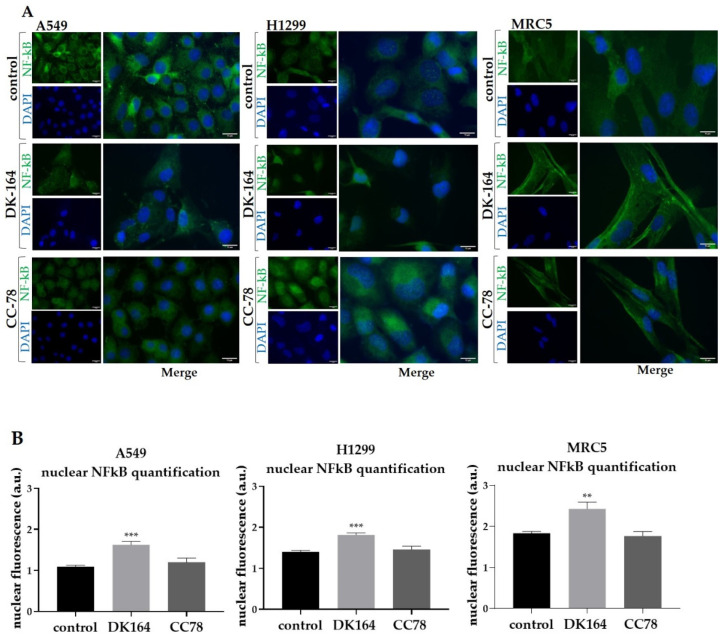
Activation and nuclear translocation of the NFkB protein in lung cells. (**A**) Representative immunofluorescent images of A549, H1299, and MRC5 control cells and cells treated with the calculated IC50 concentrations of DK-164 or CC-78 for 48 h and labelled with anti-NFkB antibody (green). DNA was co-stained with DAPI (blue). (**B**) Quantitative determination of the intensity of the NFkB was performed by CellProfiler cell image analysis software (Broad Institute′s Imaging Platform, Cambridge, MA, USA) and GraphPad Prism v.8.0 (Dotmatics, San Diego, CA, USA). A *p*-value < 0.05 was considered statistically significant, ** *p* < 0.01, and *** *p* < 0.005. Quantification is based on three independent experiments with >50 cells scored for each condition. The error bars represent the standard deviation (SD). a.u.: arbitrary unit. The scale bar corresponds to 15 μm.

**Figure 8 biomedicines-10-01353-f008:**
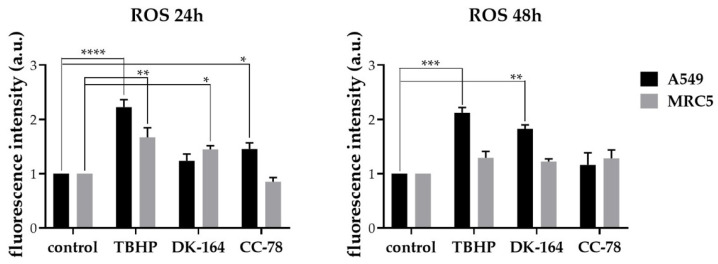
Reactive oxygen species (ROS) were induced by DK-164 and CC-78 in A549 cells and MRC5 cells. Cells were treated with IC25 of DK-164 and CC-78 for 24 h or 48 h, and ROS
were measured using the DCF-DA microtiter plate assay. *Tert*-butyl-hydroperoxide (TBHP) was used as a positive control. Data represent the mean from three independent experiments, where each experiment was performed in triplicate. Probability values were considered significant at the * *p* < 0.05, ** *p* < 0.01, *** *p* < 0.005, and **** *p* ≤ 0.001. The error bars represent the standard deviation (SD). a.u.: arbitrary unit.

**Figure 9 biomedicines-10-01353-f009:**
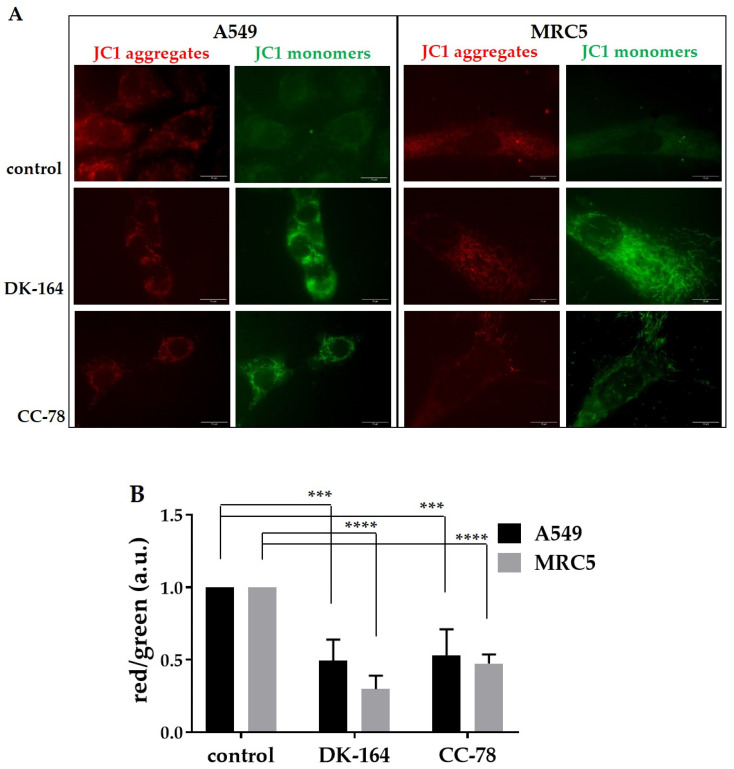
Detection of mitochondrial membrane potential by JC-1 staining. (**A**) Mitochondrial membrane depolarisation after treatment of A549 and MRC5 cells with IC25 of DK-164 and CC-78. Representative immunofluorescent images of control and treated cells for 24 h were visualised by JC-1 staining. The scale bar corresponds to 15 μm. (**B**) Quantification of the ratio of red signal (JC1 aggregates) to green signal (JC1 monomers), normalised to the intensity of the signal in non-treated control cells. Probability values were considered significant at the *** *p* < 0.005, and **** *p* < 0.001. The error bars represent the standard deviation (SD). a.u.: arbitrary unit. Bar graphs represent the means of three independent experiments.

**Table 1 biomedicines-10-01353-t001:** Calculated IC50 [µM] values for DK-164, CC-78, and the chemotherapeutics cisplatin and tamoxifen. Data show mean ± SD from at least *n* = 3 independent experiments.

	*H1299*	*A549*	*MRC5*
*DK-164*	22 ± 5.2	18.7 ± 2.3	130.7 ± 8.6
*CC-78*	19 ± 3.7	12.4 ± 2.8	10 ± 0.8
*cisplatin*	28.6 ± 4.6	19.7 ± 3.5	15.4 ± 1.9
*tamoxifen*	35.5 ± 5.8	17.8 ± 2.1	26.9 ± 2.5

## Data Availability

The data presented in this study are available on request from the corresponding author.

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
