# Peer review of "In Vitro Anticancer Activity of Two Ferrocene-Containing Camphor Sulfonamides as Promising Agents against Lung Cancer Cells"

_biomedicines, 2022, doi:10.3390/biomedicines10061353_

Round 1
Reviewer 1 Report
The manuscript by Schröder and colleagues reports the cytotoxic and antineoplastic potential of the two ferrocene containing camphor sulfonamides on two lung cancer cell lines, A549 and H1299 and one non-cancerous lung cell line, MRC5. The authors analyzed the cell cycle, apoptosis and autophagy to provide the first information about the mechanism of action of both substances on lung cells. Furthermore, the authors tested the effect of the compounds on the cellular localisation of NFkB and the tumour suppressor factor p53, critical players in essential cellular processes and for the induction of reactive oxygen species (ROS). Finally, they have also analyzed the effect of compounds on mitochondrial membrane potential.
This is a well-written manuscript. The methods are adequately described and the data supports the authors conclusions.
The only request from my side is the quantification of the results obtained from the mitochondrial membrane potential assay based on the JC-1 probe. Although the representative images are very clear, it would be nice to have a quantification of the signal resembling aggregates vs the signal resembling monomers under control or treated conditions in the different cell lines. Other than this minor revision, I think this study is acceptable for publication.
Author Response
Response: We thank reviewer #1 for positive feedback and valuable comments. We have been able to address the recommendation, and we prepared an additional panel (B) for Figure 9, presenting the quantification of the ratio of red signal (JC1-aggregates)/green signal (JC1-monomers). Please find a pdf of our new Figure 9.

Reviewer 2 Report
The manuscript entitled “In vitro anticancer activity of two ferrocene-containing camphor sulfonamides as promising agents against lung cancer cells” by Schröder et al reported testing the cytotoxic potential of two ferrocene derivatives with different residues. The manuscript is well-written and the results obtained are interesting. Some comments must be addressed before publication.
1- Line 15; Mention the name of the two ferrocene derivatives between hyphens before the word with.
2- Authors must carefully review the manuscript and mention the full description of the abbreviations at their first presence.
3- Keywords; the phrase “lung cancer cell line” please replace with “lung cancer” to be more concise.
4- Line 162-163; please remove the redundancy.
5- Authors started the Results section the chemistry without giving any information about it in the methods. Please add a paragraph about chemistry in the methods section.
6- Please clarify the word “octuplicates” that has been mentioned in line 103.
7- Authors did not mention how the IC50 was calculated.
8- Line 139; please clarify what is meant by “PBS for 2×5 min”.
9- In the Results section, 3.2. Solubility; authors did not mention any data about the procedure followed to study the solubility and the actual solubility of the two compounds.
10- Line 456, authors should clarify that gossypol is a phenolic aldehyde compound present in cotton plant. This will be clearer to the reader.
11- Line 431; what is meant by “ferrocifenes”. Please clarify.
Author Response
Response: We consider all reviewer’s #2 recommendations very reasoned and made corrections accordingly:
- We added the names of the two ferrocene derivatives at the pointed place.
- We double-checked the whole text and described the abbreviations at their first presence.
- We replaced the phrase “lung cancer cell line” with “lung cancer” in the keywords.
- We removed the redundancy “n=3”.
- All details concerning chemistry, synthetic methods, analyses and spectra of synthesised compounds are presented in Supplementary Data. The necessary short text is added to section 2.1.
- Concerning remarks #6, #7 and #8, we thoroughly edited sections 2.3 and 2.5 in Material and Methods.
- We determined the solubility of the compounds by direct measurements. While compound DK-164 is not soluble in both water and 0.01 M aqueous citric acid, compound CC-78 is very soluble in 0.01 M aqueous citric acid – more than 10 mg/ml. This is much more than concentrations, used for the determination of cytotoxicity. The solutions are stable for days and precipitates or crystals were not observed.
- We extended the description of gossypol, including an additional citation. We hope this clarifies the issue.
- We extended the description of ferrocifenes, including two new sentences and an additional citation. We hope this clarifies the issue.
Reviewer 3 Report
The work systematically studied the cytotoxic effect of designed antitumor drugs DK-164 and CC-78 against non-small lung cancer cell lines A549 and H1299 as well as their mechanism of toxicity. Authors have demonstrated designed ferrocene derivatives DK-164 and CC-78 compounds with remarkable selectivity toward cancer cells. There are many grammatical and sentence errors in the article, and the language organization needs to be improved. For these reasons, I conclude that the paper should undergo major revisions.
1. More emphasis on the finding (numerical values) and its implication may be mentioned in the abstract like the IC50 values of DK-164 and CC-78 compared to standard drugs.
2. Authors have mentioned the calculated IC50 concentrations of DK-164 or CC-78 for 24 and 48 hours in materials and Method section 2.3. But in Table 1. Calculated IC50 [μM] values the IC50 are calculated at what time 24 or 48 hours. Clarify???
3. In the Study of apoptosis, the Authors have mentioned that used calculated IC50 concentrations of each compound and analyzed by flow cytometry. Provide the data in numerical values. Also mention in the legend of Figure 3.
4. In the Study of apoptosis. Mention the % of live, early apoptotic, late apoptotic, and necrotic cells in a Table for better understanding.
5. There are many grammatical and sentence errors in the article, and the language organization needs to be improved
Author Response
Response: We thank the reviewer for the comments. We have addressed the concerns that Reviewer #3 pointed out.
- We thoroughly edited section 2.4 in Material and Methods to make our proceeding clearer and more exact.
- IC50 values were calculated in GraphPad Sigma v.8 software using the “log of concentration vs normalised response (variable slope)” algorithm. The corresponding IC50 values are presented in Table 1- they are calculated after 72 hours of incubation of the cells with the tested compounds. In all following experiments, cells were treated with the appropriate means of IC50 concentrations listed in Table 1.
- To clarify the text of Figure 3, we added a remark to Table 1.
- In fact, Figure 4 is based on a table with the substantial percentages of live, early apoptotic, late apoptotic, and necrotic cells. We provide this Table to the reviewer as an attached file.
- To satisfy this remark, we gave the revised version of our manuscript for check by a native English-speaking colleague.

Reviewer 4 Report
The manuscript “In vitro anticancer activity of two ferrocene-containing camphor sulfonamides as promising agents against lung cancer cells” develops two ferrocene derivatives against lung cancer cells. The authors demonstrated meticulous characterizations of such compounds (cytotoxicity, apoptosis, autophagy, activation of ROS). The work is interesting and precedent, although it is preliminary.
I think it is acceptable after some revision, taking into account the following points.
Minor points:
1. Page 8. Section “3.4. Autophagy in lung cells induced by treatment with DK-164 and CC-78” quantified the number of cells with enhanced autophagy (LC3 positive), based on percentage. From the same existing data set, could the authors also extract information about LC3 protein level, based on intensity?
2. Figure 4. It would be clearer to include y-axis’s axis label, for example “cell (%)”, instead of leaving it blank.
3. Page 6 and Page 8 used the same sub section number 3.4. “3.4. Study of apoptosis in lung cells treated with DK-164 and CC-78” and “3.4. Autophagy in lung cells induced by treatment with DK-164 and CC-78”
Author Response
Response: We thank reviewer #4 for positive feedback and valuable comments.
- We agree with the recommendation to quantify protein levels of LC3, but we don’t think that signal intensity from fluorescence microscopy is the right indicator. Quantification of LC3-positive puncta is considered a gold-standard assay for assessing the numbers of autophagosomes in cells, so we chose this approach.
- Figure 4 was corrected according to your recommendations and sent here as an attached file.
- We corrected the numbering of the sections.

Round 2
Reviewer 3 Report
All the queries have been addressed and may be accepted for publication